# Is It Safe to Stay at Home? Parents’ Perceptions of Child Home Injuries during the COVID-19 Lockdown

**DOI:** 10.3390/healthcare10102056

**Published:** 2022-10-17

**Authors:** Eirini Papachristou, Savas Deftereos, Panagoula Oikonomou, Konstantina Bekiaridou, Soultana Foutzitzi, Ioannis Gogoulis, Xenophon Sinopidis, Konstantinos Romanidis, Alexandra Tsaroucha, Katerina Kambouri

**Affiliations:** 1Department of Pediatric Surgery, Democritus University of Thrace, 68100 Alexandroupolis, Greece; 2Department of Radiology, Alexandroupolis University Hospital, Democritus University of Thrace, 68100 Alexandroupolis, Greece; 3Department of Experimental Surgery, Alexandroupolis University Hospital, Democritus University of Thrace, 68100 Alexandroupolis, Greece; 4Department of Pediatric Surgery, University of Patras, 26504 Patras, Greece; 5Department of Surgery, Alexandroupolis University Hospital, Democritus University of Thrace, 68100 Alexandroupolis, Greece

**Keywords:** child, home injury, COVID-19, parents’ perceptions, Greece, health policy

## Abstract

The COVID-19 pandemic and stay-at-home regulations have increased child home injuries. This study illustrates the type and frequency of child home injuries in Greece during the COVID-19 lockdown. Moreover, the survey reports the results on parents’ proposals regarding child injuries at home during the COVID-19 quarantine. A community-based, cross-sectional, descriptive study was conducted from November to December 2021 in Greece. Parents were asked to voluntarily complete an anonymous questionnaire, designed for the needs of the research. A statistical analysis of the data was performed using the Kolmogorov–Smirnov and Shapiro–Wilk tests for a normal distribution, a chi-squared (χ^2^) test to compare percentages among different groups and a non-parametric Mann–Whitney U test to determine the differences in Likert scale variables between two groups. A total of 130 parents with at least one child were questioned through an online questionnaire survey. Of the parents, 39.3%, stated that the number of accidents in their home increased. The most frequent accidents were injuries (49.3%). Most of the accidents occurred inside the house (75.8%) and were observed among children aged 0–4 years. A high percentage of children’s accidents was observed in rural/island areas or in the suburbs. Children who were with either their father or mother had one accident, and a higher number of accidents occurred when the children were with their grandparents, with the nanny or alone. For those parents who had difficulty supervising their child, child accidents increased compared to parents who had the ability to supervise. It was noticed that parents who knew how to provide a safe home stated that the number of accidents remained the same. Parents must organize a safer home. Authorities should educate parents on child injury prevention and provide them with financial facilities to provide a safer house.

## 1. Introduction

The Coronavirus disease (COVID-19) was characterized by the WHO as a pandemic on 11 March 2020 [1]. The COVID-19 pandemic has dramatically changed the lives of people and has affected the entire world. The authorities, aiming to mitigate the virus’s spread and to decrease pressure on health care systems, implemented unprecedented measures such as social distancing recommendations and school closures, eventually leading more than 2.6 billion people going into lockdown [2]. In May 2020, because of the pandemic, 99% of children worldwide lived under some sort of restrictions, as 60% lived in countries that had a partial or full lockdown, and 1.5 million children did not attend school [3].

On 27 February 2020, the first case of COVID-19 was confirmed in Greece. On 11 March, authorities implemented several restrictions, including the closure of schools and finally a strict general lockdown imposed on 23 March 2020, including a ban on gatherings and travel and the closure of non-essential shops, cafés, restaurants, bars, movie theaters, gyms, museums and archaeological sites. People had to work from home, and public spaces, universities, schools, day care centers and extracurricular activities were also closed. The first lockdown in Greece ended on 3 May 2020. On 7 November 2020, the country re-entered a second equally strict lockdown, which ended on 15 May 2021 [4,5].

Quarantining is a public health tool and aims to protect public health. It is essentially based on a public health contract, whereby people give up certain individual rights to protect other people in their community [6]. Unfortunately, the special needs of children during disasters and epidemics are often unrecognized [7]. Lockdown inevitably had serious consequences on the lives and well-being of children and young people, extending beyond those of direct viral infection [8]. Have we, as a society, seriously considered the effects of the quarantine and prolonged lockdowns, during the recent pandemic, on children’s health, particularly in relation to home injuries?

Unintentional childhood injury is always an important public health issue [9]. It is also one of the preventable causes of pediatric mortality and morbidity [10]. According to data from the World Health Organization, in 2012, deaths at home were among the top twenty causes of death in the 0–14 age group worldwide [11], particularly after the age of 1 year [9]. As defined by the WHO, unintentional injuries occur without any plan or intent [12]. Childhood unintentional injuries include falls, burns or fires, cuts, drowning, suffocation, being struck by/against, poisonings, electrical shocks, animal bites and insect stings, transport accidents and others [13].

In Greece, accidents are the first cause of death of children. According to the records, between 1992 and 2004, in Greece, there were 1593 involuntary deaths of children aged 0 to 14 years. A total of 874, or nearly half, were due to traffic accidents, 135 were due to drowning, 82 were due to drops, 47 were due to burns, 23 were due to poisonings and 432 were due to unspecified causes. Of the accidents, 80% occurred inside the house, and 20% occurred outside. In fact, 70% of injuries inside the house occurred in the living room and bedroom where children are during most hours, in the presence of adults [14]. Furthermore, as shown by the 2011 census data, in Greece, more than 700 children lose their lives each year due to accidents, and about 3000 are disabled. The incidence of accidental deaths is over 30% in children 1–4 years old [15].

Due to mandatory lockdowns in almost every country, school and extracurricular activity closures, social distancing policies, isolation periods, etc., parents and children were forced to spend much more time at home, resulting in an increase in child home injuries [16,17,18,19]. The characteristics of unintentional childhood injury during the COVID-19 pandemic and home confinement may be different from those before the pandemic. However, these new characteristics (the leading causes, the related social factors such as associations with age, sex and family types) during the COVID-19 pandemic remain unknown [20].

No information exists so far about these implications during the COVID-19 pandemic in Greece. Such information would be extremely important for parents as well as for governments worldwide. The purpose of the present study was to illustrate the type and frequency of child home injuries in Greece during the COVID-19 pandemic quarantine. Moreover, parents’ perceptions of child home injuries during the lockdown pandemic and how they can be prevented in a possible future lockdown were investigated.

## 2. Materials and Methods

### 2.1. Study Design and Population

This study follows a non-experimental correlational design. In brief, this means that data are collected in order to interpret if certain things tend to co-occur and if they are related to each other. The time dimension of the study is cross-sectional. The study was conducted between November and December 2021 and was completed by a convenient sample of 130 participants. The method of data collection was performed via a self-administered, web-based questionnaire, which was sent to the relevant subjects widely by parenting websites aimed at parents of children aged 0–14 [21]. The questionnaire consisted of 11 demographic questions related to the parents’ demographics, 3 questions related to the children’s demographics and 18 questions related to childhood accidents. Moreover, it was addressed exclusively to parents who resided permanently in Greece. The questionnaire had to be completed by only one of the two parents, and it took about 15 min to complete. The questionnaire was tested extensively through a pilot study with 15 volunteers and was adjusted accordingly. Moreover, all respondents received the same standardized questions. The purpose of this was to prevent respondents from interpreting the questions differently. Most of the data collected from the questionnaire were operationalized using a different response format (yes/no, Likert-type anchored and closed-ended questions). This made it possible to process the data quantitatively; therefore, the developed hypotheses could be tested with the use of statistical tools.

### 2.2. Statistical Analysis

Quantitative variables are presented as mean (SD) and median (IQR) values. Qualitative variables are expressed as absolute and relative frequencies (N, %). The normal distribution of the data was assessed using the Kolmogorov–Smirnov and Shapiro–Wilk tests. The chi-squared (χ^2^) test was used to compare percentages among different groups. A non-parametric Mann–Whitney U test was used to determine the differences in Likert scale variables between two groups. All statistical analyses were performed using IBM SPSS Statistics version 25.0. Differences were considered significant if *p* < 0.05 (two-tailed).

### 2.3. Ethics Approval

The study was conducted according to the guidelines of the Declaration of Helsinki [22], and the original protocol was approved by the Medical Ethics Committee of the Democritus University of Thrace, Greece; (approval number 12216/71-21 October 2021). The participants were recruited using social media, and—prior to answering the questionnaire—all participants provided informed consent after being informed about the aims and details of the study. It was highlighted in the call for participation in the present study that participation was voluntary, and participants could withdraw at any moment.

## 3. Results

The socio-demographic characteristics of the sample are shown in Table 1. According to the socio-demographic characteristics, the mean (SD) ages of fathers and mothers were 43.82 (6.12) and 40.73 (5.08) years, respectively. The majority of the participants were married (89.3%, N = 125) with 46.4% (N = 65) having two children and 40.0% (N = 56) having one child. Regarding the parents’ level of education, similar percentages were observed in both parents, as most participants were university graduates or holders of a master’s degree (22.1%, 22.9%: father, 32.9%, 27.1%: mother). A total of 42.9% (N = 60) of the participants lived in city centers, followed by those living in urban areas (30.0%, N = 42). It was also observed that the occupational status of most of the fathers did not change during the quarantine/lockdown pandemic (63.6%, N = 89), in contrast to mothers who 52.1% (N = 73) stated that there was a certain change. Among parents who stated that their occupational status had changed, most responded that they started working remotely (35.0%: father, 49.4%: mother). Finally, 47.1% (N = 66) stated that the number of accidents in their home remained the same, 39.3% (N = 55) stated that it increased and only 3 (2.1%) participants noticed a decrease in accidents.

The socio-demographic characteristics of the children and information concerning children’s accidents are shown in Table 2. Τhe percentages of boys and girls were 55.0% (N = 77) and 43.6% (N = 61), respectively. The percentages of children in groups of 0–4 years, 5–9 years and 10–14 years were 31.4% (N = 44), 43.6% (N = 61) and 23.6% (N = 33), respectively. Of the children, 67.9% (N = 95) were first-born children. A total of 52 (37.1%) households faced a serious or minor accident, 38.5% (N = 20) stated that there was one accident, 28.8% (N = 15) reported two accidents and 28.8% (N = 15) reported more than three accidents. A majority of the accidents occurred inside the house (75.8%, N = 47) and, more specifically, in the living room (49.3%, N = 34). The most frequent accidents were injuries (49.3%, N = 35), followed by falls (29.6%, N = 21). Finally, parents considered that careful supervision of their children (34.1%, N = 47) as well as the creation of a safe environment at home (15.2%, N = 21) should be taken seriously given the future confinement of children at home.

Table 3 represents comparisons of parents’ opinions based on whether the number of childhood accidents increased or remained the same. It was observed that, for those parents who had difficulty constantly supervising their child at home, child accidents increased during the pandemic compared to the parents who had the ability to supervise, with the latter reporting that accidents remained the same (*p* < 0.001). In addition, there was an increase in childhood accidents in families where parents agreed more that quarantine was responsible for the increase in accidents (*p* < 0.001). Furthermore, those parents who believed that accidents in the home increased were more likely to agree that childhood accidents are random events that cannot be controlled, compared to parents who responded that there was no increase in accidents, who believed that they can be controlled (*p* = 0.037). It was noticed that parents who knew how to provide a safe home stated that the number of accidents remained the same compared to parents who did not know how to create a safe home environment, therefore facing an increase in accidents (*p* = 0.006). We noticed an increase in accidents in households, where parents believed that human error or the underestimation of risk were responsible for such accidents. On the other hand, parents who did not believe in human error stated that there was no increase in accidents (*p =* 0.015). Finally, children’s accidents increased in households where parents considered that their child would be safer at school than at home (*p* < 0.001).

It was also examined whether the age of children was related to the frequency of child accidents. Statistically significant differences were observed (*p* < 0.001), with most accidents occurring among children aged 0–4 years, followed by the age category of 5–9 years and finally 10–14 years. In addition, a significantly high percentage of children’s accidents was observed in rural/island areas as well as in the suburbs compared to the city centers and urban areas, where the majority of parents answered that no child accidents occurred during quarantine/lockdown due to the COVID-19 pandemic (*p* = 0.026) (Table 4).

A Pearson X^2^ test was also conducted to examine whether the number of accidents were related to the family member who was with the child at the time of the accident. It was noticed that most of the children who were with either the father or the mother had one accident, and a significantly higher number of accidents occurred when the children were with their grandparents, siblings, a relative, a stranger, a nanny or even alone (*p* = 0.051) (Table 5).

Moreover, the women who declared that their occupational status incurred a change and had to work from home supported that there was an increase in children’s accidents; however, this was not statistically significant (*p* = 0.599) (Table 6).

## 4. Discussion

The social restrictions according to the COVID-19 pandemic has changed our lives, especially everyday routines, which were renamed to be ‘stay home, stay safe’. The issue of ‘safe’ was clear and related to viral infection and the reduction of its spread. However, a not-so-small number of researchers were asking questions about the other side of the problem. Would the COVID-19 restrictions have an impact on a different ‘safety’, the safety from injury and, particularly, children’s safety from injury?

Childhood injury has always been one of the most serious public health problems that requires urgent attention. The special characteristics of childhood explain the increased frequency of child injuries. However, many injuries and their consequences, are preventable. Parents and caregivers need to utilize different self-protection measures and provide a safe environment in order to reduce the risk of accidents and injuries [23]. During the COVID-19 pandemic, two very strict lockdowns were implemented in Greece, leading, among other negative consequences, to an increase in child home injuries.

Among our representative sample, 39.3% of the parents stated that the number of accidents in their home increased, and 37.1% of households faced a serious or minor accident. Similar results were found in another study in Greece and in other countries in which there was a comparison of indoor accidents in children in the 3-month period during the lockdown versus the same period in 2019 [17,18,24,25]. The most frequent accidents in our study were injuries (49.3%), followed by falls (29.6%), which is similar to another study [26]. It is surprising that there were accidents even when the parents were present, which is a fact that may indicate low parental supervision (26), and a significantly higher number of accidents occurred when the children were with their grandparents, siblings, a relative, a stranger, a nanny or even alone. It should be noted that younger children were often left alone at home during the COVID-19 pandemic, as the parents would either be busy with their own distance working or would help their eldest child during distance learning. It is also known that the lack of or lower levels of adult supervision has been linked with a higher risk of unintentional injuries, of more severe injuries in young children and of other negative outcomes [27].

According to the study results, most accidents were observed among children aged 0–4 years. This result is similar to other study results, which sustain that the prevalence of accidents increase during the first 2 years of age, reaching a peak at the age of 3–4 years [28]. Older children during quarantine stayed at home in front of screens without physical activity and, consequently, without injuries [29]. On the other hand, preschool children were forced to be inside the house, most of the time without careful supervision, but they kept their playfulness, which led them to inside injuries [30,31,32]. A significantly high percentage of children’s accidents was observed in rural/island areas as well as in the suburbs, likely because it is more difficult to supervise children in such conditions compared to supervising the narrow space of an apartment or because children in rural areas are less exposed to social isolation and have a lack of contact with peers, so the accidents were not limited [33].

It was also observed that the occupational status of the mothers had changed. Most responded that they started working remotely. During the pandemic, working mothers especially have faced unique challenges of simultaneously juggling employment and increased domestic responsibilities during the absence of stable childcare and schooling options [34]. Additionally, other research during the pandemic showed that the amount of time families spent in front of screens increased, so parents could not properly supervise their small children at home [29]. Therefore, creating a safe home environment for children was not an easy task. We would also say that, in such conditions, there can be no proper supervision of children. This is alarming if we consider that parents’ practices and attitudes are the main components for ensuring a safe environment for the physical and mental development of children [26,28].

It was observed that, for those parents who had difficulty constantly supervising their children at home during the COVID-19 pandemic lockdown, child accidents increased during the pandemic compared to parents who had the ability to supervise, with the latter reporting that accidents remained the same. Therefore, as confirmed by previous research, parental behavior, home environments and home arrangements affect the incidence of injuries within it [26]. It has also been referred by other research that, during the lockdown, the division of childcare changed. There was increased involvement of fathers in this task, which is a fact that may explain their difficulty in supervision due to reduced experience (before the quarantine, most fathers used to spend time with their children in leisurely outdoor activities) [35].

In addition, there was an increase in childhood accidents in the families of our study, where parents agreed more that the quarantine was responsible for the increase in child home accidents. These families were very likely to have experienced significant changes in their daily lives during the pandemic, for example, changes concerning the distance work of parents or even unemployment, as well as financial, health and psychological problems [29]. Moreover, quarantine is a psychologically stressful experience by itself, so it is not uncommon for parents to have increased nervousness. Therefore, it could be difficult for them to organize a calm and safe environment for their children at home [7].

According to the study results, parents considered that careful supervision of their children as well as the creation of a safe environment at home should be taken seriously given the future confinement of children at home during a possible next wave of the pandemic. It is important that, even though home accident prevention is multifactorial, and, for this reason, difficult to achieve [26], parents in our study recognize the necessity of their role in reducing child home injuries.

One of the major strengths of the current study is the participation of a diverse group of families across Greece during the COVID-19 pandemic. In addition, this study is the first to be conducted in Greece concerning children’s injuries during this period. Despite the strengths, certain limitations need to be highlighted. Firstly, the study does not include parents who lack sufficient knowledge of the Greek language. Additionally, the majority of participants lived in city centers and in urban areas. Furthermore, data were collected through an online questionnaire, which may have contributed to difficulty of access for parents with lower educational qualifications or who lack information on technology resources. Some might also wonder if the elapsed time between the end of the lockdown and the completion of the questionnaires influenced families’ responses due to forgetfulness and distortion of memories. However, although the lockdown period in Greece was ended when the study began, many restrictions (only vaccinated indoors, wearing a protective mask all around and testing twice a week for work and school) on previously free habits continued to exist for a long time due to increased cases of coronavirus infections in the community; therefore, the answers were not affected by forgetfulness.

## 5. Conclusions

The present research is the first to be carried out in Greece concerning child home injuries during the COVID-19 pandemic. According to our findings, parents faced many difficulties during the quarantine. The lockdown period could not help them to supervise their children at home and made them vulnerable to home accidents. Parents realized the importance of their role in shaping a safe home, especially in times of confinement. These findings could have serious implications, not only for parents but also for the authorities. The authorities should raise awareness and educate parents on child injury prevention issues, but they should also provide them with financial help in order to design a safe house for children. It is also suggested that national governments or administrators should be very careful when deciding to implement lockdown measures when a public crisis appears. The effects of these measures on the health and well-being of citizens and especially children should be seriously considered. We strongly believe that both governments and families should develop comprehensive planning for the prevention of unintentional childhood injury in exceptional circumstances, such as the quarantine period. Further research according to this project will examine the way parents try to create a safe environment at home for their children and the correlation of large families with pediatric home accidents during the lockdown.

## Figures and Tables

**Table 1 healthcare-10-02056-t001:** Demographic characteristics.

	N	%
**Father’s age**		
Mean (SD)	43.82 (6.12)	
Min–Max	26–61	
**Mother’s age**		
Mean (SD)	40.73 (5.08)	
Min–Max	27–54	
**Family status**		
Married	125	89.3
Divorced	9	6.4
Unmarried	1	0.7
Separated	1	0.7
Living with a partner	3	2.1
Widower	1	0.7
**Number of children in the family**		
1	56	40.0
2	65	46.4
3	16	11.4
4+	3	2.1
**Father’s education level**		
Elementary/High School graduate	8	5.7
Lyceum graduate	20	14.3
Graduate of Technical High School	11	7.9
School of the Workforce Employment Agency		
Graduate of Vocational Training Institute	13	9.3
Graduate of Technological Educational Institute	19	13.6
Graduate of University	31	22.1
Master’s degree	32	22.9
PhD degree	6	4.3
**Mother’s education level**		
Elementary/High School graduate	2	1.4
Lyceum graduate	8	5.7
Graduate of Technical High School	5	3.6
School of the Workforce Employment Agency		
Graduate of Vocational Training Institute	16	11.4
Graduate of Technological Educational Institute	19	13.6
Graduate of University	46	32.9
Master’s degree	38	27.1
PhD degree	6	4.3
**Residence**		
City centers	60	42.9
Urban area	42	30.0
Suburbs	25	17.9
Rural area	9	6.4
Island area	4	2.9
**Was there any change in the father’s occupational status during the quarantine/lockdown pandemic?**		
Yes	50	35.7
No	89	63.6
Not answered	1	0.7
**Which of the following changes occurred for the father?**		
I was unemployed.	3	5.0
My business is closed.	3	5.0
I had a suspended employment contract.	6	10.0
I got a special permit.	5	8.3
I worked remotely from home.	21	35.0
Working hours increased.	12	20.0
Working hours decreased.	7	11.7
Other	3	5.0
**Was there any change in the mother’s occupational status during the quarantine/lockdown pandemic?**		
Yes	73	52.1
No	67	47.9
Which of the following changes occurred for the mother?		
I was unemployed.	3	3.4
I had a suspended employment contract.	5	5.7
I got a special permit.	8	9.2
I worked remotely from home.	43	49.4
Working hours increased.	18	20.7
Working hours decreased.	5	5.7
Other	5	5.7
**Do you think that during the pandemic and lockdown child accidents at home:**		
Increased	55	39.3
Decreased	3	2.1
Remained the same	66	47.1
I do not know/Not answered	16	11.4

**Table 2 healthcare-10-02056-t002:** Socio-demographic characteristics of the children and information concerning children’s accidents.

	N	%
**What is the gender of your child?**		
Boys	77	55.0
Girls	61	43.6
Do not want to specify	2	1.4
**What is the age of your child?**		
0–4	44	31.4
5–9	61	43.6
10–14	33	23.6
Not answered	2	1.4
**Number of children in the family**		
1	95	67.9
2	35	25.0
3+	10	7.1
**Has your child had a SERIOUS or MINOR accident/accidents at home or in the surrounding area during quarantine/lockdown due to the COVID-19 pandemic?**		
Yes	52	37.1
No	88	62.9
**If yes, how many accidents did you have?**		
1	20	38.5
2	15	28.8
3+	15	28.8
Not answered	2	3.8
**Where did the accident/accidents happen?**		
Inside the house	47	75.8
On the terrace	5	8.1
In the yard	9	14.5
Not answered	1	1.6
**If the accident(s) occurred in the house, at what point/places in the house did it occur?**		
Kitchen	15	21.7
Bathroom	3	4.3
Bedroom	16	23.2
Living room	34	49.3
Not answered	1	1.4
**In which category does the accident/accidents that happened belong?**		
Falls (cot, stroller, crib, changing table, walker, relax, balcony, etc.)	21	29.6
Ιnhalation of objects (nuts, toys, coins, batteries, buttons, etc.)	2	2.8
Ingestion of a foreign body (nuts, toys, coins, batteries, buttons, etc.)	1	1.4
Suffocation	1	1.4
Burns (hot water, flame, chemicals, etc.)	8	11.3
Electrocution	1	1.4
Injuries (toys, furniture corners, glass surfaces, tools, animals, weapons, etc.)	35	49.3
Poisonings (drugs, cleaners, cosmetics, cigarettes, pesticides, etc.)	1	1.4
Other	1	1.4
**In general, based on your experience as a parent from the quarantine due to the pandemic, what suggestions would you like to make, such that, in the case of future confinement of children at home, their stay in it is safer?**		
I do not know/I do not answer	19	13.8
Creating a safe environment at home	21	15.2
More careful supervision of children	47	34.1
Informing parents and children about the prevention of child accidents at home	9	6.5
Open schools/kindergartens/extracurricular activities	12	8.7
Less remote working; distance parenting	7	5.1
No quarantine or confinement in the future	6	4.3
Creative work at home	8	5.8
Activities for children outdoors	9	6.5

**Table 3 healthcare-10-02056-t003:** Comparison between the question, “Do you think that, during the pandemic and lockdown, child accidents at home:” and the parents’ perceptions about children’s accidents.

	Do You Think That, during the Pandemic and Lockdown, Child Accidents at Home:			
	Increased	Remained the Same	Z	*p*
Given the circumstances, the closure of schools, extracurricular activities and the confinement of children at home for all this time was the right decision.	2 (1–4)	3 (1–4)	−0.519	0.604
I had difficulty constantly supervising my child at home.	4 (3–5)	3 (2–4)	−3.512	**<0.001**
Quarantine at home increased my child’s chances of having an accident.	4 (3–5)	2 (1–3)	−6.600	**<0.001**
Children’s accidents at home are random events that cannot be controlled.	3 (2–4)	3 (3–4)	−2.089	**0.037**
Children’s schools and extracurricular activities should not have been closed completely.	4 (3–5)	4 (3–5)	−1.099	0.272
Children’s accidents can be prevented.	3 (3–4)	3 (3–4)	−0.806	0.420
I do not know how to provide a safe house to protect my child from accidents.	2 (2–3)	2 (1–2)	−2.758	**0.006**
Children’s accidents at home are due to human error or underestimating risk.	4 (3–4)	3 (3–4)	−2.422	**0.015**
My child is safer at school than at home.	3 (2–4)	2 (1–3)	−3.837	**<0.001**

Note: Bold values are with the statistical significance.

**Table 4 healthcare-10-02056-t004:** Factors associated with children’s accidents.

	Did Your Child Have a Specific Accident, Whether Serious or Minor, at Home or in the Immediate Environment during the Quarantine?			
	Yes	No	X^2^	*p*
**What is the age of your child?**				
0–4	26 (50%)	18 (20.9%)	17.838	**<0.001**
5–9	22 (42.3%)	39 (45.3%)		
10–14	4 (7.7%)	29 (33.7%)		
**Residence**			9.251	**0.026**
City center	17 (32.7%)	43 (48.9%)		
Urban area	14 (26.9%)	28 (31.8%)		
Suburbs	12 (23.1%)	13 (14.8%)		
Rural area/Island area	9 (17.3%)	4 (4.5%)		

Note: Bold values are with the statistical significance.

**Table 5 healthcare-10-02056-t005:** Factors related to the number of child accidents.

	Number of Accidents				
	1	2	3+	X^2^	*p*
When the accidents occurred, the child was at home with:				18.262	**0.051**
Father/Mother	18 (78.3%)	11 (44%)	14 (56%)		
Siblings or other children	4 (17.4%)	8 (32%)	4 (16%)		
Grandparents	1 (4.3%)	2 (8%)	5 (20%)		
Relative/Stranger/Nanny/Other	0 (0%)	2 (8%)	1 (4%)		
Alone	0 (0%)	2 (8%)	1 (4%)		

Note: Bold values are with the statistical significance.

**Table 6 healthcare-10-02056-t006:** Comparison of the increase in accidents in relation to the mother’s change in occupational status.

	Do You Think That, during the Pandemic and Lockdown, Child Accidents at Home:			
	Increased	Remained the Same	X^2^	*p*
Mothers worked remotely from home:			0.276	0.599
Yes	18 (64.3%)	22 (57.9%)		
No	10 (35.7%)	16 (42.1%)		

## Data Availability

The datasets generated during and/or analyzed during the current study are available from the corresponding author on reasonable request.

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
