# Peer review of "Is It Safe to Stay at Home? Parents’ Perceptions of Child Home Injuries during the COVID-19 Lockdown"

_healthcare, 2022, doi:10.3390/healthcare10102056_

Round 1

Reviewer 1 Report

Dear authors,

The manuscript titled "Is it Safe to Stay at Home? Parents’ Perceptions of Child Home 2 Injuries During COVID-19 lockdown" is related to child home injuries during the pandemic and explained the child home injuries have increased during the pandemic in Greece. The manuscript is short and concise. It is easy to read and follow the manuscript. 

I have some recommendations to enhance the quality of the manuscript. 

- It should add to the frequency of child home injuries in general in Greece.

- Which injuries could be defined as child home injuries? It could be detailed. 

- There should be references in the second paragraph of the introduction part.

- In the method section, there was no information about measurement. Is there any standardized measurement or the authors have prepared with the literature? How many questions and any validity and reliability of the form?

- In the results, the sociodemographic characteristics could be given in Table. There were lots of numbers and percentages that were difficult to follow.

- The discussion is well written and clear.

- The limitations of this study should be added. For example; the number of participants was very limited for a community-based survey. 

Have success in your studies. 

Reviewer 2 Report

The article “Is it Safe to Stay at Home? Parents’ Perceptions of Child Home Injuries During COVID-19 lockdown” considers that it does not meet various requirements that it should observe as a result of a scientific investigation process. The document is too short and lacks several fundamental sections in this class of research reports, in addition to the fact that the subject seems to be an overly elementary descriptive matter. Among the elements identified with observations are the following: (a) The sampling used is not indicated, it is only presented that there were 130 parents surveyed, but it is not known how they were selected, what is the total population, what sociodemographic characteristics they observe to be selected in the sample, etc.; (2) the keywords are only 3, they are too generic and imprecise; (3) the document lacks a theoretical-referential framework, which would allow the reader and the researcher himself to identify antecedents in this kind of studies; (4) although the objective is obvious, a concrete definition based on general and specific objectives is not presented within the methodology; (5) I don't think it is quite correct to collect qualitative data through the use of electronic surveys, which regularly work with quantitative data; (6) indicates that the data collection instrument has 41 items, however the analysis seems to only analyze three of them. There is much imprecision in this aspect of the research; (7) the discussion section should relate the results of the research to current scientific literature; (8) the conclusion does not express results arising from the research itself and is only based on brief general comments; (9) the references that support the research work are limited and its format should also be reviewed because it seems that it does not adhere to the regulations of the journal itself.

Reviewer 3 Report

The present study deals with a topic of great importance for the care and attention of children. Specifically, it analyzes the type and frequency of child home injuries in Greece during the COVID-19 lockdown. Undoubtedly, the study of this issue will help to improve the training of families in this area and, ultimately, to modify parental educational styles. Nevertheless, the manuscript presents some aspects that need to be considered and reviewed.

-The abstract must be written without headings. In addition, it would be advisable to allude to the type of analysis performed.

-The definition of the state of the art in the Introduction section is excessively brief and scanty. In this sense, it would be interesting to investigate, for example, what has happened in other countries, which would reduce the localist nature of the study. Likewise, it would be advisable to delve deeper into the importance of childhood unintentional injury, since it is only tangentially mentioned. Therefore, the authors are actively encouraged to expand the theoretical foundation of the study.

-Along the same lines, it is recommended to indicate the duration of the second period of confinement, since it is only stated that it began on November 7, 2020 (lines 52-53).

-On the other hand, it is suggested that the bibliographic search be expanded, since the number of sources consulted (17) is too few for this area of knowledge.

-Data were collected between November and December 2021, although the second period of lockdown in Greece was in November 2020. Could the elapsed year have influenced the families’ responses? Was the effect of forgetfulness and distortion of memories on the responses taken into account? How were attempts made to control for these factors? If they were not contemplated, it would be advisable to consider them as limitations of the research.

-Why was Google Forms, and not other tools, used for the online design of the questionnaire? What were the characteristics of the questionnaire (topics addressed, type of questions, etc.)? Was it validated before its application? If so, in what way? What social networks were used for its dissemination? What was the procedure followed? Etc. Basic information is missing in the Materials and method section.

-For the presentation of the sociodemographic results (lines 97-122), the use of tables is suggested. This would facilitate the understanding of the information.

-Statistical symbols, such as the significance level, should appear in italics.

Lines 197-201 state: “It was also observed that the occupational status of the mothers had changed. Most responded that they started working remotely. During the pandemic especially working mothers have faced unique challenges in simultaneously juggling employment and increased domestic responsibilities during the absence of stable childcare and schooling options”. It would be necessary to allude to the lack of co-responsibility in the distribution of domestic and childcare tasks.

-The authors are encouraged to include future lines of research.

-Finally, it is recommended that the conclusions drawn from the study be expanded, as they are too brief.

Round 2

Reviewer 2 Report

Substantial improvements to the document are observed. It is recommended that tables 1 and 2, given their length, be submitted as annexes if the journal allows it.

Reviewer 3 Report

The authors have made all the suggested modifications. However, if tables with sociodemographic data are included, it is necessary to delete the information from the text. This avoids duplication of information.
